# Tumor Androgen Receptor Protein Level Is Positively Associated with a Better Overall Survival in Melanoma Patients

**DOI:** 10.3390/genes14020345

**Published:** 2023-01-28

**Authors:** Nupur Singh, Jude Khatib, Chi-Yang Chiu, Jianjian Lin, Tejesh Surender Patel, Feng Liu-Smith

**Affiliations:** 1College of Medicine, University of Tennessee Health Science Center, Memphis, TN 38103, USA; 2Department of Dermatology, College of Medicine, University of Tennessee Health Science Center, Memphis, TN 38103, USA; 3Department of Preventive Medicine, College of Medicine, University of Tennessee Health Science Center, Memphis, TN 38103, USA

**Keywords:** melanoma, androgen receptor, protein expression, melanoma survival, cancer

## Abstract

Androgen receptor (AR) is expressed in numerous tissues and serves important biologic functions in skin, prostate, immune, cardiovascular, and neural systems, alongside sexual development. Several studies have associated AR expression and patient survival in various cancers, yet there are limited studies examining the relationship between AR expression and cutaneous melanoma. This study used genomics and proteomics data from The Cancer Proteome Atlas (TCPA) and The Cancer Genome Atlas (TCGA), with 470 cutaneous melanoma patient data points. Cox regression analyses evaluated the association between AR protein level with overall survival and revealed that a higher level of AR protein was positively associated with a better overall survival (OS) (*p* = 0.003). When stratified by sex, the AR association with OS was only significant for both sexes. The multivariate Cox models with justifications of sex, age of diagnosis, stage of disease, and Breslow depth of the tumor confirmed the AR-OS association in all patients. However, the significance of AR was lost when ulceration was included in the model. When stratified by sex, the multivariate Cox models indicated significant role of AR in OS of female patients but not in males. AR-associated genes were identified and enrichment analysis revealed shared and distinct gene network in male and female patients. Furthermore, AR was found significantly associated with OS in RAS mutant subtypes of melanoma but not in BRAF, NF1, or triple-wild type subtypes of melanoma. Our study may provide insight into the well-known female survival advantage in melanoma patients.

## 1. Introduction

Melanoma incidence continues to increase worldwide; in the US, it has increased by 320% since 1975 [1]. Research on how sex hormones and their receptors impact melanoma have not resulted in solid conclusions. Androgen receptor (AR), for example, was recently reported to exhibit effects of promoting cell proliferation, melanoma metastasis, and drug resistance in melanoma cells and mouse models [2,3,4]. While molecular studies and mouse models have provided much interesting information, we are interested in investigating whether AR was differentially expressed in melanoma tumors from men and women, and whether the tumor AR levels are associated with patient overall survival (OS). This study shall shed insight into a long-observed phenomena, i.e., the female survival advantage of melanoma patients [5,6].

As a male sex hormone receptor, the gene *AR* is located on the X chromosome [7]. Two androgenic hormones that are able to bind to AR include testosterone (T), and its metabolite dihydrotestosterone (DHT), and they are active in human skin in endocrine and paracrine manner [8,9,10]. AR and these hormones exert their genomic effects via induction of transcriptional activities, and non-genomics activity through signal transduction, both of which are best studied in human prostate cancer [11,12,13].

Sex differences in cancer incidence have also been documented in several cancers, [14]. For instance, higher incidence rates of lung, liver, stomach, esophageal, and bladder malignancies alongside cutaneous melanoma are found in males compared to females [15,16,17,18]. Aside from lifestyle, the characterization of the molecular differences in cancer between male and female malignancies highlights the sex-based variations of gene expression on a molecular level [19]. Nonetheless, there remains a lack of complete understanding of what role AR signaling plays in most hormone-independent cancers alongside cutaneous melanoma.

Current literature has explored possible pathways into AR’s effects, both harmful and protective, on cutaneous melanoma development. One proposed mechanism explains how AR and the protein Early Growth Response 1 (EGR1) increase melanoma proliferation through coordinated transcriptional regulation of several growth-regulatory genes, including the repression of EGR1-mediated transcriptional activation of p21Waf1/Cip1, a known tumor suppressor gene [20]. Another mechanism suggesting melanoma progression includes altering the miRNA-539-3p/USP13 signaling to reduce de-ubiquitination of MITF protein, increasing MITF degradation, and allowing further invasion [4]. Other mechanisms of AR’s role in cancer risks have been proposed to provide a potential protective effect, specifically cancer-associated fibroblast (CAF) activation. Decreased AR expression in primary human dermal fibroblasts (HDFs) derived from multiple individuals led to early steps of CAF activation. The discovered mechanism includes the development of a complex in which AR combines with CSL/RBP-Jκ to normally repress the transcription of key CAF effector gene [21].

The conflicting findings of AR’s protective or harmful role in melanoma progression shows the complexity of several convergent mechanisms likely implicated in melanoma’s AR dependency. In this paper, we use The Cancer Genome Atlas (TCGA) and The Cancer Proteome Atlas (TCPA) to evaluate the relationship between *AR* gene and AR protein expression in human cutaneous melanoma, and their association with OS in patients. Genomic network was further explored in an attempt to understand the sex-differentiated roles of AR in patient overall survival.

## 2. Materials and Methods

### 2.1. The Source of Data

The data source used for all analyses (TCGA-SKCM) were obtained from The Cancer Genome Atlas (TCGA), with mRNA sequencing (RNA-Seq) data downloaded from Broad Firehose GDAC (http://gdac.broadinstitute.org/, accessed on 14 June 2022), and proteomics data from The Cancer Proteome Atlas (TCPA) (https://tcpaportal.org/, accessed on 14 June 2022). The RNA-Seq data were retrieved as RSEM (RNA-Seq by Expectation-Maximization) and Z scores [22]. Patient ID, sex, age of diagnosis, follow-up time, and survival status were also downloaded from the Broad GDAC site. The database contained 480 tumors from 471 patients with cutaneous melanoma. Protein expression data were available for 355 tumors. If patient duplicates were encountered, the data for metastatic tumor was selected and the primary tumor data were discarded. Tumor stages are grouped into early (Stage I and Stage II) or late stage (Stage III and Stage VI) or used as denoted in the dataset as Stage 0–4. 

### 2.2. Statistical Methods

All statistics were analyzed using Stata 17. Linear regression was used to examine the association of mRNA and protein levels in tumor samples. AR levels (mRNA or protein) were compared between sex by Student *t*-test and/or rank-sum test. Cox regression analyses were performed to evaluate the association between AR protein level with overall survival. The AR-high and AR-low groups were defined by the median of AR protein level (−0.718). The regression model was further stratified by sex or adjusted to age, tumor stage (early and late), Breslow depth, and ulceration status of the tumors. Sex was also used as an adjusting co-variable in the overall model. The overall survival was defined as the period from date of diagnosis until death from any cause. Significance levels are set at 0.05 (two-sided) for all analysis.

### 2.3. Gene Network Analysis

AR-co-expressed genes (based on RNA-Seq) were extracted from the cBioportal website (https://www.cbioportal.org/, accessed on 14 June 2022) using sex-stratified patient information. The AR-co-expressed genes were processed using adjusted q values of 0.05, followed by cutoff value of Spearman’s coefficient of 0.3 [23]. A set of genes that are uniquely associated with AR in male and female tumors were identified and then subjected to a functional enrichment analysis using g:profiler web-based analysis tools (https://biit.cs.ut.ee/gprofiler/, accessed on 14 June 2022).

## 3. Results

### 3.1. The Sex Difference of AR Gene Expression at mRNA and Protein Level

The TCGA SKCM dataset was downloaded from the Broad Institute Firehose website. The baseline characterizations of patients are listed in Table 1. The protein quantification data are available for 353 patients and the mRNA data are available for all 471 patients. The mRNA level of AR was compared between tumors from male and female sources using log transformed RSEM. A total of 21 female tumors and 34 male tumors did not show detectable levels of mRNA (RSEM = 0), but they were also included in the analysis. Student *t*-test showed no sex difference in mean of log transformed RSEM values (*p* = 0.10). However, when the protein levels were used for sex comparison, tumors from females (N = 144) showed a significant lower level of AR protein than those from males (N = 208) (*p* = 0.0099) (Table 2).

We then investigated whether tumor mRNA and protein levels of AR are positively associated. In fact, a linear regression model between AR protein and log-transformed RSEM data showed significant positive association of AR at mRNA and protein level (*p* < 0.0001 for all samples together, *p* = 0.003 for females and *p* < 0.0001 for males) (Table 2, Appendix A).

### 3.2. Tumor AR Protein Levels Are Positively Associated with Patient Overall Survival

The sex difference in survival is well known for melanoma. In order to examine whether AR plays a role in such sex difference, melanoma patients are grouped by their tumor AR protein levels. “AR-high” group of patients have tumor AR levels greater than median AR (−0.718) for the entire cohort, while “AR-low” group of patients have tumor AR levels lower than the median AR. Our initial Kaplan–Meier survival analysis suggested that higher AR levels were associated with better OS (Figure 1), and this result seemed true for all patients (log rank test *p* = 0.0025), for male patients (*p* = 0.046), or for female patients (*p* = 0.0107) (Figure 1a–c). Cox regression analysis (simple variate analysis) further revealed that higher AR levels were significantly associated with better OS in females (*p* = 0.012) or for all patients (*p* = 0.003), and the significance level was reduced to 0.047 (*p* value) in male patients (Table 3). Cox proportional assumption was tested based on the Schoenfeld residues, and a *p* value of 0.24 was returned, indicating Cox analysis was a proper method for survival analysis for this dataset, which is consistent with our previous report [24]. In this dataset, sex alone was not a significant determinant for overall survival (Cox regression HR = 1.14, *p* = 0.39) (Table A1).

In the multivariate analysis, sex was used as a co-variable, and the AR association with overall survival was additionally adjusted by age of diagnosis, stage of disease (either stage 0 to stage 4 or early and late stages, as described in Section 2) (Table 3, Model 1–3). The association of AR with overall survival stayed significant after adjusting to these factors. When the presence of ulceration was added in the multivariate analysis, the association lost its significance (HR = 0.80, *p* = 0.29) (Table 3, Model 4). Interestingly, ulceration alone was significantly associated with overall survival (HR = 1.8, *p* = 0.001), and the significance remained after adjusting to age of diagnosis, stage of disease, and sex (HR = 1.46, *p* = 0.04).

These results may suggest an impact of AR on the ulceration status. However, tumors with or without ulceration did not show significant difference in the AR protein levels (*p* = 0.23 in Student *t*-test).

Another important prognostic factor for melanoma survival is Breslow depth. We grouped Breslow depth according to the AJCC TNM staging standards (1 = <1 mm; 2 = 1.01–2 mm; 3 = 2.01–4 mm; 4 = >4 mm) and included this variable in our multivariate COX analysis. The AR association with overall survival remained significant when the result was adjusted to Breslow depth, along with other factors (HR = 0.59, *p* = 0.008) (Table 3, Model 5).

### 3.3. The Sex Difference in the AR Association with OS

Table 4 shows that the AR protein level is not significantly associated with overall survival in men, even though the high AR is significantly associated with overall survival in women. Sex was then used as a stratification variable and the multivariate Cox models in male and female patients were analyzed separately, with age, stage, ulceration status, and Breslow depth as adjusting co-variables. AR levels were not associated with men’s OS in any of the models, but they are associated with women’s overall survival in all models except for Model 4 where ulceration was justified.

Since testosterone levels are known to change with men’s age, we also examined whether AR levels in tumors were different in older versus younger patients (≤50 vs. >50 years). A Student *t*-test was used to evaluate the AR protein levels, and no difference in means was found (*p* = 0.89 for men and 0.13 for women).

### 3.4. The Differential AR Gene Network in Tumors from Men and Women

In order to understand how AR expression levels are associated with patient overall survival in women but not in men, the TGCA SKCM mRNA data were used to extract the AR co-expressed genes using the online tool from the cBioportal website. The entire genome was included and the co-expressed genes were identified using a cutoff q value of q < 0.05. 6413 genes from men and 3384 genes from women were retained for further comparison. When the Spearman’s co-efficient for AR-association was set at ρ > 0.34, then 75 genes in women and 202 genes in men were retained for further comparison. Among these genes, 44 were unique for women, 171 were unique for men (Table A2), and 31 were shared by tumors from both sexes (Table A3, Appendix A). The 10 most significant genes for each sex are included in Table 5.

The 44 and 171 genes identified in the female and male tumors, respectively, are subjected to enrichment analysis using an integrated web-based tool termed g:profiler (https://biit.cs.ut.ee/gprofiler/gost, accessed on 14 June 2022). Genes were ordered All significant enrichments for females and partial of that for males are listed in Table 6. For female tumors, AR is significantly associated with GO:MF (molecular function), GO:CC (cellular component), and TF (transcription factor) functions. For male tumors, AR is significantly associated with a wide range of functions, including 99 GO:BP (biological process), 19 GO:CC, 10 GO:MF, 8 TF, 1 Reactome (Neurophilin interactions with VEGF and VEGFR), and 4 WP (Wikipathways). Only the top three significant functions in each category are shown in the table. 

The shared 31 genes in male and female tumors were used for the same profiling analysis, 18 enriched functions were identified, which are listed in Table A4.

### 3.5. The Role of AR in Overall Survival in Four Melanoma Subtypes

The TCGA melanoma team classified this cohort of patients into four distinct subtypes with distinct somatic mutations in the tumors [25]. We obtained the classification information at patient level from their supplemental tables. A total of 316 patients were included in the analysis, but due to some tumors lacking AR protein data, only 230 patients were included in the survival analysis. Very interestingly, only in the RAS (mainly NRAS, but also including several mutants in KRAS and HRAS) mutants, did AR show significant association with overall survival (*p* = 0.013) (Table 7). The significance remained after adjusting to age of diagnosis and stage of disease (*p* = 0.047). When only sex is adjusted, the significant also remained (*p* = 0.022), but it was reduced to borderline (*p* = 0.057) when both sex and age are included.

## 4. Discussion

The findings of this study suggest that a higher level of tumor AR protein is positively associated with a better overall survival in cutaneous melanoma patients, which remains true after adjusting to age of diagnosis, stage of disease, sex of patients, and Breslow depth of the tumors. However, when patients are stratified by sex, the significant association was found only in female patients, but not in male patients, even though sex itself is not significantly associated with overall survival in this dataset. Additionally, when ulceration status is included in the model, the significance of AR association with OS was lost, suggesting that ulceration is still the most effective prognostic factor for melanoma OS. A statistical test of an interaction of AR with ulceration status revealed only borderline significance (*p* = 0.10, not shown in results). Nevertheless, our finding is significant, as this is one of the first studies to show an association of tumor AR level with overall survival in melanoma patients.

A previous report suggested an opposite role of AR in melanoma patient survival, i.e., higher AR was associated with worse survival [4]. That report did not specify melanoma subtype. The samples were collected in China, while the melanoma subtype in China is different than that in US—Chinese melanoma cases are mostly acral melanoma, which are distinct in oncogenic causes and pathological pathways than the US cases, which are mostly superficial spreading melanoma [26,27].

For melanoma, similar to many other cancer types, females in general show a survival advantage even after adjusting to many other prognostic factors. The underlying mechanism may be multi-fold, and we have been interested in the roles of sex hormones in such situations. Sex hormones and their receptors play critical roles in many pathophysiological conditions and impact many oncogenic pathways and cellular functions. AR was recently studied in melanoma cells, with a function of promoting proliferation, tumorigenesis, metastasis, and drug resistance [2,3,4], which is opposite to our findings.

The possible explanation may be directly linked to the androgen levels as the majority function of AR is linked to locally available testosterone and dihydrotestosterone. Therefore, in most cases, we must study the function of AR/T or AR/DHT together. It is particularly important to study the sex-specific impact, as men and women are distinctly different in the circulating T or DHT levels. Our study also showed a distinct gene network in tumors from male and female patients, further strengthening the importance of sex-specific investigation. Another possible reason is related to how to interpret the data. In one study, loss of AR led to more DNA damage [2], suggesting that AR played a protective role in genome integrity. When this occurs in normal melanocytes, one would expect AR serves as a tumor suppressor, as it was found in a subset of breast cancer [28]. This is also what our study suggests.

It is also noticeable that AR plays distinct functions in the male and female tumors, with shared functions in both sexes. The enriched functions are much broader in male tumors, indicating male-biased significance of AR. Since AR is involved in many more gene networks in males, the ability of these functions to maintain a relative cellular balance may be strengthened, which may help to explain why AR in men did not show a significant association with overall survival.

The weakness of this study is that we used only the TCGA data, with no replicating dataset. Therefore, this study requires further validation in a different patient cohort. As noted in one of our previous study [24], the patient sex did not show a significant association with OS, which is not the usual case for melanoma patients. That is the limitation of the patient cohort as well, and requires further replication.

## 5. Conclusions

The overall conclusion of this study is that tumor AR protein levels are associated with better OS in female patients, and not in male patients. We have also identified shared and distinct AR-associated gene networks in male and female tumors, which suggests AR exhibits common function in all tumors, and also exhibits distinct function in tumors from male and female patients. This study is the first to include data from a large database source with over 350 data points from melanoma patients’ tumors. Most of previous studies of AR in melanoma suggested that AR promoted tumor proliferation, metastasis, and drug resistance. Our study suggests that the role of AR should be considered in sex-specific manner, and in females, AR could be protective. Further investigation on these shared and distinct functions of AR in melanoma patients will help us to develop precise treatment strategies.

## Figures and Tables

**Figure 1 genes-14-00345-f001:**
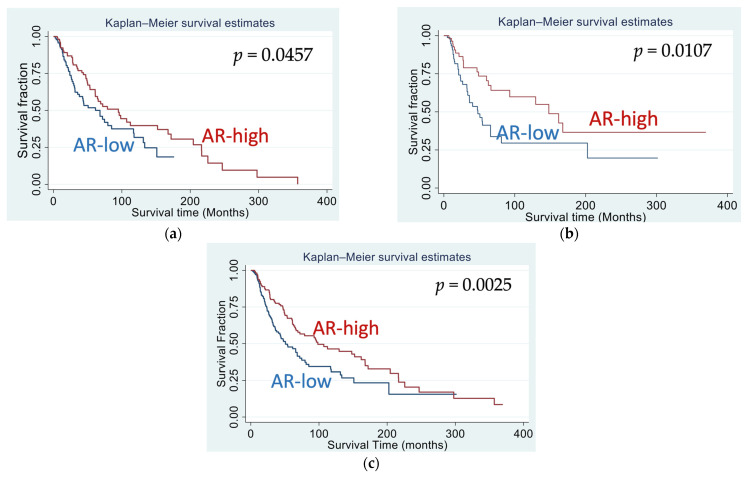
Kaplan–Meier survival curves for patients with high and low AR protein levels. (**a**) For both male and female patients; (**b**) for female patients; (**c**) for male patients (*p* values are derived from log rank test).

**Table 1 genes-14-00345-t001:** Baseline characterization of patients.

	Female	Male	Missing *	Total
Number of patients	180	290	1 (sex)	471
Number of tumors	183	296	1 (sex)	480
Tumors with available AR RPPA data	144	208	1 (sex)	353
Tumors with available AR mRNA data	180	289	1 (mRNA)	471
Number of primary tumors	45	64	0	109
Number of metastatic tumors	138	232	1 (sex)	371
Age at diagnosis (years)	58.5 ± 1.2	58.0 ± 0.9	9 (age)	58.2 ± 0.7
Stage of disease				
stage 0	2	5		7
stage 1	25	52		77
stage 2	61	93		154
stage 3	67	104		171
stage 4	8	15		23
missing	17	21	1 (sex)	39
total	180	290		471
Stage of disease **				
early	88	150		238
late	75	119		194
missing	17	21	1 (sex)	39
Ulceration				
no	57	89		146
yes	67	100		167
missing	56	101		158
Breslow depth				
<1.0 mm	20	36		56
1.0–2.0 mm	27	53		80
2.0–4.0 mm	33	44		77
>4.0 mm	60	83		143
missing	40	74	1 (sex)	115

* missing means the number of patients missing the corresponding data, e.g., first row, 1 (sex) means 1 patient missing sex information. ** stages 0–2 are defined as early stage, while stages 3–4 are late stage.

**Table 2 genes-14-00345-t002:** The sex difference in AR gene expression.

	mRNA			Protein		
Sex	Female	Male	Total	Female	Male	Total
N	180	289	469	144	208	352
Mean	−0.077	−0.053	−0.062	−0.744	−0.665	−0.691
Std.err.	0.065	0.048	0.039	0.023	0.019	0.015
Median	−0.347	−0.288	−0.312	−0.762	−0.695	−0.718
95% CI	−0.205	−0.148	−0.138	−0.789	−0.703	−0.727
	0.051	0.041	0.013	−0.698	−0.627	−0.668
*p* value (sex difference)	0.77			**0.0099**		
mRNA vs. protein (linear regression)	Female: coefficient: 0.10 ± 0.03, *p* = 0.003					
	Male: coefficient: 0.11 ± 0.02, *p* < 0.0001					
	All: coefficient: 0.11 ± 0.017, *p* < 0.0001					

**Table 3 genes-14-00345-t003:** AR is significantly associated with overall survival in melanoma patients.

Analysis	Patients/Model	HR	95% CI		*p* Value *	Variable(s) Included
Simple variate	Female	0.49	0.28	0.86	0.012	AR
	Male	0.66	0.44	0.99	0.047	AR
	All	0.61	0.44	0.84	0.003	AR
Multivariate	Model 1	0.65	0.47	0.90	0.009	AR, sex, age
	Model 2	0.67	0.48	0.95	0.025	AR, sex, age, stage (0–4)
	Model 3	0.68	0.48	0.96	0.029	AR, sex, age, stage (early, late)
	Model 4	0.80	0.54	1.20	0.29	AR, sex, age, stage (early, late), ulceration
	Model 5	0.59	0.40	0.87	0.008	AR, sex, age, stage (early, late), Breslow depth (4 category)

*, *p* value: for AR.

**Table 4 genes-14-00345-t004:** AR protein level for survival between female and male sexes.

		Female				Male			
	Variables	HR	[95% Conf. Interval]		*p* Value	HR	[95% Conf. Interval]		*p* Value
Model 1	AR	0.51	0.29	0.90	**0.021**	0.70	0.47	1.06	0.092
	age	1.03	1.02	1.05	0	1.02	1.01	1.04	0.003
Model 2	AR	0.49	0.27	0.89	**0.02**	0.80	0.52	1.24	0.328
	age	1.03	1.01	1.05	0.001	1.02	1.00	1.03	0.028
	stage (0–4)	1.27	0.91	1.79	0.165	1.46	1.14	1.87	0.003
Model 3	AR	0.48	0.26	0.87	**0.016**	0.84	0.54	1.31	0.443
	age	1.03	1.01	1.05	0	1.02	1.00	1.03	0.017
	stage (early, late)	1.36	0.77	2.42	0.289	1.96	1.26	3.08	0.003
Model 4	AR	0.58	0.28	1.19	0.135	1.02	0.60	1.73	0.951
	age	1.03	1.01	1.05	0.009	1.02	1.00	1.04	0.106
	Stage (early, late)	1.50	0.77	2.91	0.23	2.16	1.26	3.71	0.005
	ulceration	1.28	0.64	2.58	0.489	1.60	0.91	2.81	0.1
Model 5	AR	0.49	0.25	0.96	**0.039**	0.65	0.39	1.09	0.102
	age	1.03	1.01	1.05	0.004	1.02	1.00	1.04	0.088
	Stage (early, late)	1.58	0.86	2.90	0.143	1.38	0.79	2.41	0.258
	Breslow Depth	1.27	0.92	1.74	0.149	1.71	1.28	2.28	0

**Table 5 genes-14-00345-t005:** Top 10 most significant sex-specific AR co-expressed genes in tumors.

Gene	Spearman’s Coefficient	*p* Value	q Value	Sex	Approved Gene Name	HGNC ID	Location
KMT2A	0.42	4.3 × 10^−9^	0.000025	F	lysine methyltransferase 2A	HGNC:7132	11q23.3
NECTIN3	0.41	8.8 × 10^−9^	0.000025	F	nectin cell adhesion molecule 3	HGNC:17664	3q13.13
ROR1	0.41	1.4 × 10^−8^	0.000029	F	receptor tyrosine kinase like orphan receptor 1	HGNC:10256	1p31.3
MACF1	0.41	1.6 × 10^−8^	0.000029	F	microtubule actin crosslinking factor 1	HGNC:13664	1p34.3
CBL	0.39	5.5 × 10^−8^	0.000065	F	Cbl proto-oncogene	HGNC:1541	11q23.3
AKAP2	0.39	7 × 10^−8^	0.000078	F	A-kinase anchoring protein 2	HGNC:372	9q31.3
KERA	0.39	7.5 × 10^−8^	0.00008	F	keratocan	HGNC:6309	12q21.33
PRDM10	0.39	8.3 × 10^−8^	0.000082	F	PR/SET domain 10	HGNC:13995	11q24.3
MAML2	0.38	9.9 × 10^−8^	0.00009	F	mastermind like transcriptional coactivator 2	HGNC:16259	11q21
ZFP91	0.38	1 × 10^−7^	0.00009	F	ZFP91 zinc finger protein, atypical E3 ubiquitin ligase	HGNC:14983	11q12.1
SLIT2	0.49	7.6 × 10^−19^	7.7 × 10^−15^	M	slit guidance ligand 2	HGNC:11086	4p15.31
ITGA8	0.45	1.1 × 10^−15^	4.5 × 10^−12^	M	integrin subunit α 8	HGNC:6144	10p13
PREX2	0.45	1.1 × 10^−15^	4.5 × 10^−12^	M	phosphatidylinositol-3,4,5-trisphosphate dependent Rac exchange factor 2	HGNC:22950	8q13.2
MARCHF8	0.43	1 × 10^−14^	2.4 × 10^−11^	M	membrane associated ring-CH-type finger 8	HGNC:23356	10q11.21-q11.22
RALGAPA2	0.43	1.3 × 10^−14^	2.6 × 10^−11^	M	Ral GTPase activating protein catalytic subunit α 2	HGNC:16207	20p11.23
ZDHHC15	0.43	2.5 × 10^−14^	4.2 × 10^−11^	M	zinc finger DHHC-type palmitoyltransferase 15	HGNC:20342	Xq13.3
IL6ST	0.42	4 × 10^−14^	6 × 10^−11^	M	interleukin 6 cytokine family signal transducer	HGNC:6021	5q11.2
PCSK5	0.42	8.5 × 10^−14^	1 × 10^−10^	M	proprotein convertase subtilisin/kexin type 5	HGNC:8747	9q21.13
MAN1A1	0.42	9.3 × 10^−14^	1 × 10^−10^	M	mannosidase α class 1A member 1	HGNC:6821	6q22.31
ASXL3	0.42	1.2 × 10^−13^	1.3 × 10^−10^	M	ASXL transcriptional regulator 3	HGNC:29357	18q12.1

**Table 6 genes-14-00345-t006:** The sex-specific AR-associated enrichment of gene function.

Sex	Source	Term_Id	Adjusted_*p*_Value	Term_Size	Query_Size	Intersection_Size	Term_Name
Female	GO:MF	GO:0042800	0.028832	18	14	2	histone methyltransferase activity (H3-K4 specific)
	GO:MF	GO:0106363	0.042024	2	1	1	protein-cysteine methyltransferase activity
	GO:CC	GO:0043296	0.009441	154	27	4	apical junction complex
	TF	TF:M09984_1	0.007033	5696	43	27	Factor: MAZ; motif: GGGGGAGGGGGNGRGRRRGNRG; match class: 1
	TF	TF:M12654_1	0.032391	44	3	2	Factor: PRDM15; motif: NYCCRNTCCRGGTTTTSC; match class: 1
	TF	TF:M09834_1	0.032799	2950	39	17	Factor: ZNF148; motif: NNNNNNCCNNCCCCTCCCCCACCCN; match class: 1
Male	GO:MF	GO:0046872	5.32 × 10^−6^	4271	131	57	metal ion binding
	GO:MF	GO:0005509	5.9 × 10^−6^	726	130	21	calcium ion binding
	GO:MF	GO:0043169	1.2 × 10^−5^	4364	131	57	cation binding
	GO:BP	GO:0048731	1.65 × 10^−11^	4369	163	78	system development
	GO:BP	GO:0048856	6.64 × 10^−11^	5836	163	91	anatomical structure development
	GO:BP	GO:0007155	2.2 × 10^−10^	1521	167	43	cell adhesion
	GO:CC	GO:0005887	3.25 × 10^−10^	1649	156	41	integral component of plasma membrane
	GO:CC	GO:0031226	3.56 × 10^−10^	1731	156	42	intrinsic component of plasma membrane
	GO:CC	GO:0071944	3.05 × 10^−8^	6270	160	85	cell periphery
	REAC	REAC:R-HSA-194306	0.005273	4	15	2	Neurophilin interactions with VEGF and VEGFR
	WP	WP:WP4823	0.004028	44	11	3	Genes controlling nephrogenesis
	WP	WP:WP3943	0.004081	6	11	2	Robo4 and VEGF signaling pathways crosstalk
	WP	WP:WP5065	0.005193	5	15	2	SARS-CoV-2 altering angiogenesis via NRP1
	TF	TF:M00695_1	3.82 × 10^−8^	7194	169	104	Factor: ETF; motif: GVGGMGG; match class: 1
	TF	TF:M12345_1	0.00052	1735	74	22	Factor: Zbtb37; motif: NYACCGCRNTCACCGCR; match class: 1
	TF	TF:M01199	0.002376	8683	169	105	Factor: RNF96; motif: BCCCGCRGCC

**Table 7 genes-14-00345-t007:** Role of AR in overall survival in four subtypes of melanoma.

	HR	[95% Conf. Interval]		*p* Value	N **
BRAF_Hotspot_Mutants	0.62	0.20	1.15	0.336	106
RAS_Hotspot_Mutants	0.44	0.23	0.84	**0.013 ***	67
NF1_Any_Mutants	0.73	0.26	2.04	0.551	25
Triple_WT	0.97	0.39	2.45	0.950	32

* *p* value for RAS subtype = 0.047 after adjusting to age and stage of patients. ** N: number of patients in each subtype included in survival analysis.

## Data Availability

All data used in this study are publicly available; the sites are listed in the manuscript.

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
