# Peer review of "Tumor Androgen Receptor Protein Level Is Positively Associated with a Better Overall Survival in Melanoma Patients"

_genes, 2023, doi:10.3390/genes14020345_

Round 1

Reviewer 1 Report

Summary
In this article, the authors investigate the association between AR gene expression, protein level and survival in publicly available melanoma datasets. They show that AR protein levels are positively associated with a better overall survival, especially in females, even after accounting for multiple covariates. They acknowledge that the significance is lost when ulceration is included in the model, possibly due to a collinearity between ulceration and AR protein levels. Finally, they identify AR-associated gene networks in males and females. This study analyses in depth, for the first time, the association between AR protein levels and survival in melanoma and its interrelation with patient sex, correcting for most of the relevant covariates. The methodologies and results are quite well described.

General comments
The main clarification that I would like to ask to the authors concerns the methodology of the Cox regression analysis. The public availability of the data and metadata allowed me to attempt to reproduce the authors' main analyses using R and the function coxph of the 'survival' package. I did obtain the same results as in Tables 1 and 2, especially concerning the sex differences in AR protein expression. However, the hazard ratios and p-values for AR, reported in Tables 3 and 4, for the univariate and multivariate (models 1-5) Cox regressions were different, in my analyses. In particular, AR did not reach significance in any of the multivariate models despite having the same input data (as I obtain the same summary statistics as in Table 2), and despite obtaining the same HR and p-values for the other covariates (age, ...) presented in Tables A1 and A2. I would thus kindly ask the authors to double-check their code and results, and to provide additional details on which Stata function they used for the Cox regression, and which parameters. It seems to me important to clarify this discrepancy since this is the main claim of the paper.

A second point I would like to raise concerns AR gene networks. As TCGA data are from bulk tumors, they contain reads from a variety of cell types. Could the authors discriminate whether the correlated genes are expressed in the same cell types (tumor cells?) or in different ones? To this end, the authors might analyse some publicly available single cell datasets such as Tirosh et al., or others. They could, in this way, (1) test whether AR is expressed in cancer cells and (2) test the coexpression of AR with other genes in single cells from male and female patients.

Third, TCGA provided a genomic classification into 4 subtypes (BRAF-, RAS-, NF1- mutant and triple-wild type). It would be interesting to test whether the survival association with AR remains significant after accounting for the genomic subtype as well, or possibly if, after stratifying, this association is found only in some but not all of these subtypes.

Specific comments
- The Cox regression assumes proportional hazard ratios. Could the authors test if the variables they are including in the various models match this assumption? They could analyse Schoenfeld residuals or use appropriate statistical tests

- It is a bit unclear what is described in lines 119-121. Why were z-scores used for the test? I recommend to use only the log RSEM values, possibly adding pseudocounts, or alternatively using the non-logged values but with a non-parametric test such as Mann-Whitney

- I would possibly move the biggest tables to supplementary files to improve the readability of the paper 

Author Response

Thank you for taking your time to review our paper and give us good feedback. By the following points we believe we have addressed all your comments:

  1. In Table 2, it would be much better and visually inviting if a dot plot could be presented to show the correlation of mRNA and protein level correlations of AR in each patient sample. Could be in the supplementary/appendix

            Response: We have included a dot plot into the appendix as the new supplemental Figure S1.

  1. In Table 6, would authors organize the table in a better way e.g. clearly show the name of genes and its corresponding attributes. In the current format it is sometimes hard to tell one gene name from another.

            Response: Yes, we added boxed lines in the table so it is easier to see. We did the same for Table 5 as well.

  1. In Table 6, it would be better if the co-expressed genes were sequenced in regard to the p-value/co-efficiency rather than alphabetical sequence.

            Response: Yes. We changed this to keeping the female and male separations but updating the table starting with lowest p value within the male and female categories. This rearranged the table to have the most significant genes with smallest p values presented first.

  1. Table A2 is good, better organize Table 6 in a consistent manner as in Table A2. Still need to address the sequence issue. 

            Response: We changed Table 6 to match Table A2. The table was also re-sorted to order genes with smallest p values.

  1. Would authors find another similar dataset to prove similar findings of AR expression correlation with OS? A simple analysis would be good enough just to justify the central conclusion of this study. There is still caveats that AR/OS correlation only exists in this one special dataset.

Response: This is a very good point, we are currently looking for other validating dataset. That may take additional work. We addressed this caveat in the discussion.

Reviewer 2 Report

As a biochemist in the field of cancer biology, we long for good molecular targets of cancer research to reveal the prognostic and therapeutic value. The manuscript by Singh et al showed a potential to fulfill such scientific needs. The function and significance of AR in cancer progression is a being constantly debated and one would expect AR expression level to promote cancer progression, but this study showed quite the opposite, which would piqued the interest of broader readership. Then they presented sounded statistical analysis to support their findings and showed the significance only presented in female but not male patients. Although this study has inherent limitations, I still recommend the publication of this manuscript if following issues could be well-addressed:

1. In Table 2, it would be much better and visually inviting if a dot plot could be presented to show the correlation of mRNA and protein level correlations of AR in each patient sample. Could be in the supplementary/appendix

2. In Table 6, would authors organize the table in a better way e.g. clearly show the name of genes and its corresponding attributes. In the current format it is sometimes hard to tell one gene name from another.

3. In Table 6, it would be better if the co-expressed genes were sequenced in regard to the p-value/co-efficiency rather than alphabetical sequence.

4. Table A2 is good, better organize Table 6 in a consistent manner as in Table A2. Still need to address the sequence issue. 

5. Would authors find another similar dataset to prove similar findings of AR expression correlation with OS? A simple analysis would be good enough just to justify the central conclusion of this study. There is still caveats that AR/OS correlation only exists in this one special dataset.

Author Response

Thank you for very constructive review and suggestions. Below please find our responses to each of your comments.

Specific comments:

1)    In the Abstract in lane 20 the Authors show the p-value=0.003 for the association of AR protein levels with better overall survival. Where exactly does this statistic come from in the main part of the manuscript? I am struggling with finding it. Please comment.

            Response: The top 3 rows in Table 3 were mis-copied from original analysis. The p value for all patient in the simple variate analysis should be 0.003 instead of 0.01 shown in the manuscript. We corrected all three rows and have looked carefully throughout all tables in the manuscript to ensure no more such mistakes.

2)    Line 32-33 – What do the Authors mean exactly by this sentence? Do they mean that the incidence is growing? Please make the sentence clear.

            Response: We changed this sentence in an attempt to make it more clear:”Melanoma incidence continues to increase world-wide; in US it has increased 320% since 1975”

3)    Line 37-39 – This is a false statement. All of the cited studies investigated cutaneous melanoma in the context of AR expression. Authors should describe what is exactly the novelty of their study, and the exact gap in knowledge addressed in this study having in mind that this topic was extensively investigated before.

            Response: we have re-written this sentence: “While molecular studies and mouse models have provided much interesting information, we are interested in investigating whether AR was differentially expressed in melanoma tumors from men and women, and whether the tumor AR levels are associated with patient overall survival (OS).”

4)    Line 119-122 – Please explain what exactly “missing patients” means in the Table. Is it the lack of RNA expression? There is a need for a more detailed explanation of the finding about “no sex differences”, which patients exactly were compared?  Because right now from how it is written in 120-121 “Those samples have Z scores..” it suggests that “those” are the patients with no mRNA expression described in the previous sentence. Please explain and change the text so that it is more clear.

            Response: We added note to Table 1 to explain “missing” column “ *missing means the number of patients missing the corresponding data. E.g, first row, 1 (sex) means 1 patient missing sex information. “  

For sex difference, we followed another reviewer’s comment and decided to show sex different in only log-transformed RSEM data but not in Z score. So this part was removed from the revision.  

5)    Line 138 - What does it mean that the protein levels were divided into halves?

            Response: We agree this sentence is not clear. Now it is rewritten: “In order to examine whether AR plays a role in such sex difference, melanoma patients are grouped by their tumor AR protein levels. “AR-high” group of patients have tumor AR levels greater than median AR (-0.718) for the entire cohort while “AR-low” group of patients have tumor AR levels lower than the median AR.”

6)  Line 141-145- Is this Cox regression analysis include the low and high AR expression analysis? What is the cut-off for the low and high expression of AR? The statistical significance should be annotated on the Kaplan-Meier survival estimates graphs.

            Response:  Simple variate Cox regression used AR as the only variable to predict survival (Table 3). As in response to Comment 5 above, AR-high and AR-low means greater or lower than median AR for the entire cohort. We added statistical p values in the graphs which are based on log rank analysis.

7) Line 180 - What is exactly the difference between this analysis and the analysis in Table 3.

            Response: The analysis in table 3 highlights the significant role of AR in different models (sex was used as a co-variable in the models) while the analysis starting from line 180 specifically compare the sex difference when each factor is considered. Here sex was used as a strata variable.

8) Line 201 - Please visualize this data with the Venn Diagram and highlight the top10 or top20 genes unique for men and women and please try to discuss this finding in more detail. Do not include all the 90 genes for men in the table, put it in supplementary and highlight only top10/top20 genes in the main text.

            Response: Top 15 genes are shown now in each sex. A new paragraph is now added to discuss the differential gene network in the discussion. A Venn diagram is added as Figure S2,

9) Line 231-232 - This is not the first study, it was previously described here https://www.nature.com/articles/onc2016330

            Response: We agree, this is not the first study. We modified our sentence to include the Oncogene paper results, which actually had an opposite conclusion. We discussed possible reasons for this discrepancy. “A previous report suggested an opposite role of AR in melanoma patient survival, i.e, higher AR was associated with worse survival [4]. That report did not specify melanoma subtype. The samples were collected in China, while the melanoma subtype in China is different than that in US – Chinese melanoma cases are mostly acral melanoma which are distinct in oncogenic causes and pathological pathways than the US cases which are mostly superficial spreading melanoma [24, 25]. “

10)  Line 240 – What do the Authors have in mind by saying “almost opposite to our findings “ , is it opposite or not?

            Response: “Almost” has been removed.

11)  Line 259 – I am struggling with understanding why the conclusion is not limited to men and women, but also includes all patient aspects. The females are the ones who contribute to the increased survival in all patients, and maybe it would be better to just state this. 

            Response: We have now changed it to “female patients”.

Major concern:

The novelty of the study is very poorly discussed and presented. There are many studies investigating the role of AR expression in the context of melanoma survival and aggressiveness. The Authors should provide a proper narrative that will show the novelty of their study.

            Response: As the reviewer pointed out, there was a previous report on AR’s role in melanoma survival, which is negative. Higher AR was associated with poor survival. Also previous cellular and molecular studies all suggest that AR plays a deleterious role in melanoma, i.e., promoting melanomagenesis, promoting metastasis, promoting drug resistance. Our conclusion is opposite to most of the published results, indicating that AR can be protective, at least in females. This is our innovation. We also pointed this out in our conclusion section: “Most of previous studies of AR in melanoma suggested that AR promoted tumor proliferation, metastasis and drug resistance. Our study suggest that the role of AR should be studied in sex-specific manner, and in females, AR could be protective.”

Reviewer 3 Report

General comments

This Original Research Paper gives insight into the correlation between Androgen Receptor expression levels and the overall survival of cutaneous melanoma patients. This study supports the well-known advantage of female survival in melanoma patients that can be associated with AR expression. The topic is important in the context of understanding melanoma epidemiology and aggressiveness. I have major concerns regarding the novelty and the discussion of the data by the Authors that need to be addressed.

Specific comments:

1)    In the Abstract in lane 20 the Authors show the p-value=0.003 for the association of AR protein levels with better overall survival. Where exactly does this statistic come from in the main part of the manuscript? I am struggling with finding it. Please comment.

2)    Line 32-33 – What do the Authors mean exactly by this sentence? Do they mean that the incidence is growing? Please make the sentence clear.

3)    Line 37-39 – This is a false statement. All of the cited studies investigated cutaneous melanoma in the context of AR expression. Authors should describe what is exactly the novelty of their study, and the exact gap in knowledge addressed in this study having in mind that this topic was extensively investigated before.

4)    Line 119-122 – Please explain what exactly “missing patients” means in the Table. Is it the lack of RNA expression? There is a need for a more detailed explanation of the finding about “no sex differences”, which patients exactly were compared?  Because right now from how it is written in 120-121 “Those samples have Z scores..” it suggests that “those” are the patients with no mRNA expression described in the previous sentence. Please explain and change the text so that it is more clear.

5)    Line 138 - What does it mean that the protein levels were divided into halves?

6)  Line 141-145- Is this Cox regression analysis include the low and high AR expression analysis? What is the cut-off for the low and high expression of AR? The statistical significance should be annotated on the Kaplan-Meier survival estimates graphs.

7) Line 180 - What is exactly the difference between this analysis and the analysis in Table 3.

8) Line 201 - Please visualize this data with the Venn Diagram and highlight the top10 or top20 genes unique for men and women and please try to discuss this finding in more detail. Do not include all the 90 genes for men in the table, put it in supplementary and highlight only top10/top20 genes in the main text.

9) Line 231-232 - This is not the first study, it was previously described here https://www.nature.com/articles/onc2016330

10)  Line 240 – What do the Authors have in mind by saying “almost opposite to our findings “ , is it opposite or not?

11)  Line 259 – I am struggling with understanding why the conclusion is not limited to men and women, but also includes all patient aspects. The females are the ones who contribute to the increased survival in all patients, and maybe it would be better to just state this. 

Major concern:

The novelty of the study is very poorly discussed and presented. There are many studies investigating the role of AR expression in the context of melanoma survival and aggressiveness. The Authors should provide a proper narrative that will show the novelty of their study.

Author Response

Thank you for your kind help. Please see below for our response. 

General comments

The main clarification that I would like to ask to the authors concerns the methodology of the Cox regression analysis. The public availability of the data and metadata allowed me to attempt to reproduce the authors' main analyses using R and the function coxph of the 'survival' package.

I did obtain the same results as in Tables 1 and 2, especially concerning the sex differences in AR protein expression. However, the hazard ratios and p-values for AR, reported in Tables 3 and 4, for the univariate and multivariate (models 1-5) Cox regressions were different, in my analyses. In particular, AR did not reach significance in any of the multivariate models despite having the same input data (as I obtain the same summary statistics as in Table 2), and despite obtaining the same HR and p-values for the other covariates (age, ...) presented in Tables A1 and A2. I would thus kindly ask the authors to double-check their code and results, and to provide additional details on which Stata function they used for the Cox regression, and which parameters. It seems to me important to clarify this discrepancy since this is the main claim of the paper.

Response: we apologize for the error in stating our grouping of patients. The median of AR was used to group AR-high and AR-low patients, not mean. We added median values in Table 2. Although the distribution of AR protein is not deviated from a normal distribution, but the mean and median still somewhat differed, we believe median is a better dividing standard. Since the reviewer mentioned the R package with different results, we performed R analysis as well using the coxph survival package, we got similar but not identical results as our Stata analysis.  The R code and results are attached in this revision for the reviewer to go through.

When addressing one of the other reviewer’s comment, we found top three rows of Table 3 was mis-copied, which are now corrected. We checked all other tables and redid our analysis in Stata to ensure no other mistake. We did find that the gene network analysis results was not based on mRNA results. We updated Tables 5 and 6.

We did perform a Cox PH assumption test based on Schoenfeld residues, the p value was 0.24, indicating no violation of the assumptions. This was added into the text.

A second point I would like to raise concerns AR gene networks. As TCGA data are from bulk tumors, they contain reads from a variety of cell types. Could the authors discriminate whether the correlated genes are expressed in the same cell types (tumor cells?) or in different ones? To this end, the authors might analyse some publicly available single cell datasets such as Tirosh et al., or others. They could, in this way, (1) test whether AR is expressed in cancer cells and (2) test the coexpression of AR with other genes in single cells from male and female patients.

Response: This is a great point and a great suggestion. For purpose 1: to validate the expression of AR in melanoma cells, we have western blot data in our laboratory to show that AR was expressed in all 8 melanoma cell lines we examined, expression levels varied but all detectable (we used AR positive prostate cancer cell line as our control). For purpose 2, this is truly an important aspect, and we intend to pursue it in future studies.

Third, TCGA provided a genomic classification into 4 subtypes (BRAF-, RAS-, NF1- mutant and triple-wild type). It would be interesting to test whether the survival association with AR remains significant after accounting for the genomic subtype as well, or possibly if, after stratifying, this association is found only in some but not all of these subtypes.

Response: Great points. We did that and added Table 7 to include the results which are very interesting, as it seems that only in RAS mutants, AR was significantly associated with overall survival. This, of course, will require future validation with larger dataset, but sure enough it is very interesting initial results.

Specific comments

- The Cox regression assumes proportional hazard ratios. Could the authors test if the variables they are including in the various models match this assumption? They could analyse Schoenfeld residuals or use appropriate statistical tests

Response: we did that and included in the revision. No violation of assumption.

- It is a bit unclear what is described in lines 119-121. Why were z-scores used for the test? I recommend to use only the log RSEM values, possibly adding pseudocounts, or alternatively using the non-logged values but with a non-parametric test such as Mann-Whitney

Response: good catch. We found this when we were trying to answer another reviewer’s question, we now only show the log rsem results.  

- I would possibly move the biggest tables to supplementary files to improve the readability of the paper

Response: we have shortened the tables to show only the most significant genes/pathways. The new tables are now not very big.

Round 2

Reviewer 1 Report

I thank the authors for replying point to point to my comments, and for providing the code run in support of their analyses.

It is now clearer after seeing their code that the Cox regression was performed on the binarized AR expression status (above or below median) instead of using AR protein levels as a continuous variable, and my analyses now match their results. This is clear reading the Results section (lines 138-142), but not so clear from the Methods section. I would thus suggest to add a sentence in the Methods section clarifying that a binarization of AR levels as performed prior to the Cox regression, which was then run on the binarized AR levels.

I congratulate the authors for this work.

Author Response

We now added a sentence in the method section: "The AR-high and AR-low groups were defined by the median of AR protein level (-0.718). "

Thank you for your validation and great suggestions.

Reviewer 3 Report

I am satisfied with the response and corrections from the Authors. 

Author Response

Thank you